# Association between intimate partner violence and male alcohol use and the receipt of perinatal care: Evidence from Nepal demographic and health survey 2011–2016

**Blessing Akombi-Inyang**[1,2]*, **Pramesh Raj Ghimire**[2,3], **Elizabeth Archibong**[2], **Emma Woolley**[4], **Husna Razee**[1]

1 School of Population Health, University of New South Wales, Sydney, Australia, 2 School of Health Sciences, Western Sydney University, Penrith, Australia, 3 Ujyalo Nepal, Ratnanagar Municipality, Nepal, 4 School of Education, Macquarie University, Sydney, Australia

* b.akombi@unsw.edu.au

## Abstract

The utilization of perinatal care services among women experiencing intimate partner violence (IPV) and male alcohol use is a major problem. Adequate and regular perinatal care is essential through the continuum of pregnancy to mitigate pregnancy and birth complications. The aim of this study is to determine the association between IPV and male alcohol use and the receipt of perinatal care in Nepal. This study used pooled data from 2011 and 2016 Nepal Demographic and Health Surveys (NDHS). A total of 3067 women who interviewed for domestic violence module and had most recent live birth 5 years prior surveys were included in the analysis. Multivariable logistic regression analysis was performed to determine the association between IPV and male alcohol use and the receipt of perinatal care. Of the total women interviewed, 22% reported physical violence, 14% emotional violence, and 11% sexual violence. Women who were exposed to physical violence were significantly more likely to report non-usage of institutional delivery [adjusted Odds Ratio (aOR) = 1.30 (95% CI: 1.01, 1.68)] and skilled delivery assistants [aOR = 1.43 (95% CI: 1.10, 1.88)]. Non-attendance of 4 or more skilled antenatal care visits was associated with a combination of alcohol use by male partner and exposure to emotional [aOR = 1.42 (95% CI: 1.01, 2.00)] and physical violence [aOR = 1.39 (95% CI: 1.03, 1.88)]. The negative association between IPV and perinatal care suggests it is essential to develop comprehensive community-based interventions which integrates IPV support services with other health services to increase the uptake of perinatal care through the continuum of pregnancy.

## Introduction

Maternal mortality refers to deaths due to complications from pregnancy or childbirth. Between year 2000 and 2017, there has been a 38% decline in global maternal mortality ratio—from 342 deaths per 100,000 live births to 211 deaths indicating an average annual reduction

**Data Availability Statement:** Our study was based on an analysis of existing datasets in the Demographic and Health Survey (DHS) repository

that are freely available online with all identifier information removed. The first author obtained authorization for the download and usage of Nepal DHS dataset from MEASURE DHS/ICF International, Rockville, MD, USA. Here are the direct links to the Nepal DHS Datasets for 2011 and 2016 respectively: https://dhsprogram.com/data/dataset/Nepal_Standard-DHS_2011.cfm?flag=0 https://dhsprogram.com/data/dataset/Nepal_Standard-DHS_2016.cfm?flag=0.

**Funding:** The author(s) received no specific funding for this work.

**Competing interests:** The authors have declared that no competing interests exist.

rate (ARR) of 2.9% [1]. Despite this significant progress, the reported global ARR in maternal mortality is less than half the 6.4% annual rate needed to meet the Sustainable Development global target of 70 maternal deaths per 100,000 live births by 2030. South Asia specifically achieved the greatest overall percentage reduction in maternal mortality rate (MMR), with a reduction of 59%—from 395 deaths per 100,000 live births to 163 deaths between 2000 and 2017 with Nepal also achieving a decline from 550 deaths per 100,000 live births to 186 deaths [1].

Most pregnancy-related complications leading to maternal mortality are preventable by quality healthcare during pregnancy and childbirth. Within the continuum of care, perinatal care is the care given to women from pregnancy through to one year after childbirth with antenatal care (ANC) and postnatal care (PNC) being care given before and after childbirth respectively. Perinatal care provides a platform for critical healthcare functions including health promotion, prevention, screening, and diagnosis of diseases. Adequate and regular perinatal care could mitigate pregnancy and birth complications, foetus and infant risk of complications as well as inform women about important steps to be taken to protect their infant and ensure a healthy pregnancy. Yet in many countries' women continue to receive inadequate perinatal care. The importance of adequate perinatal care cannot be over-emphasized as this is essential health care delivered to women with the goal of early and timely prevention, identification, and treatment of health risks that may contribute to adverse health outcomes [2]. WHO recommends a minimum of eight antenatal contacts with the first contact occurring in the first 12 weeks of gestation, and subsequently at 20, 26, 30, 34, 36, 38 and 40 weeks of gestation [2]. In Nepal, standard ANC services include at least four ANC visits (first at the fourth month, second at the sixth month, third at the eighth month, and fourth at the ninth month of pregnancy). However, the coverage of the four ANC visits within the country is not high at 69% [9], of which less than 25% of women receive good-quality ANC care [10]. Furthermore, the 2015 National Health Facility Survey reported that during the provision of perinatal care, important dimensions of ANC services such as effectiveness, efficiency, and safety were poor [12]. Research has shown that there are several factors associated with the lack of adequate and timely perinatal care [3, 4]. Experiencing Intimate Partner Violence (IPV) has been reported as one such factor associated with inadequate uptake of perinatal care [5]. IPV exposes reproductive age women to a wide range of health problems that can either directly or indirectly lead to maternal mortality and morbidity [6]. IPV affects one in three ever-partnered women worldwide in their lifetime [7]. A substantial body of evidence attests to the negative association between IPV and the uptake of adequate perinatal care, and skilled delivery care [8–11]. The World Health Organization (WHO) has identified pathways by which IPV may contribute to sexual and reproductive health outcomes. WHO notes that coercion by the perpetrator of IPV negatively affects women's autonomy, thereby limiting their ability to make decisions about when and if to seek health care during pregnancy and after childbirth. IPV also results in mental health problems such as depression and anxiety [12]. Such mental health problems reduce a woman's desire to obtain health care services and their ability to make decisions regarding their own health [12]. Though previous studies have reported an association between IPV and inadequate antenatal care [8, 9], and low utilization of skilled delivery care [10, 11], other studies have reported no evidence of an association [13, 14]. This disparity in findings may be as a result of methodological differences around definitions of IPV assessment tools and outcome measures [15, 16]. Given this inconsistency in the findings, it is important to conduct a population-based country-specific analysis to ascertain the impact of IPV on perinatal care.

In addition, research has shown that a major contributor to the occurrence of IPV is male alcohol consumption, especially at harmful and hazardous levels [17]. Alcohol impacts

cognitive and physical functioning resulting in increased aggression, reduced self-control, and a propensity to resort to violence for conflict resolution [18, 19]. While there is robust data on the association between alcohol use and IPV, it must be noted that the influence of alcohol consumption as a direct cause of IPV has been a topic of debate [16] due to the presence of additional factors such as low socio-economic status and impulsive personality [17]. These factors also could be exacerbated by frequent heavy drinking which could lead to stressful relationships which in turn increases the risk of conflict and violence. However, the role of male alcohol consumption in shaping the extent of IPV as well as its influence on the receipt of perinatal care is still largely uncertain.

In Nepal, a substantial number of reproductive age women experience IPV, which affects their health in many ways, including during and after pregnancy. A recent research reported that IPV is widespread in Nepal with about a quarter of ever-married women experiencing IPV most commonly in the form of beating, neglect, and verbal abuse [20, 21]. They identify low education, childhood exposure to parental partner violence, and alcohol misuse by husband as contributors to IPV. Patriarchal systems that perpetuate male dominance of women in Nepal makes partner violence acceptable and this acts as a barrier for women experiencing IPV to seek help [20–23].

Alcohol use in Nepal is culturally and socially acceptable and embedded as part of the religious and cultural life within some communities [24]. Alcohol is regarded as "pure offerings to God" and therefore having religious importance [24]. Although a significant proportion of the Nepalese population abstain from alcohol consumption, the prevalence of alcohol misuse have been increasing with rates of binge drinking as high as 70% [24]. Previous studies conducted in Nepal has shown that spousal violence and male alcohol use was associated with the receipt of low levels of skilled maternity care either across the pregnancy continuum or at recommended points during or after pregnancy [10]. Therefore, it is crucial to understand the association between IPV, male alcohol use and perinatal care, not only because it may increase health professionals' ability to identify women experiencing IPV, but also because of the health implications such as maternal complications and poor pregnancy outcomes in which victims of IPV face across the pregnancy continuum.

Hence, the main aim of this study is to utilise pooled data from 2011 and 2016 Nepal Demographic and Health Survey (NDHS) to determine the association between IPV and male alcohol use and the receipt of perinatal care in Nepal after controlling for potential confounding factors. Findings from this study would be useful to policy makers and public health researchers in formulating effective interventions aimed at improving the receipt of perinatal care service by reducing IPV among women of reproductive age.

## Materials and methods

This study used pooled data obtained from 2011 and 2016 NDHS [25, 26]. The surveys were implemented by New ERA under the aegis of the Ministry of Health of Nepal in conjunction with the United States Agency for International Development (USAID) and ICF Macro, Calverton, MD, USA [17]. The NDHS is a nationally representative survey which collect data on fertility, family planning and maternal and child health using standardized questionnaires, manuals, and field procedures that are comparable across countries. The 2011 and 2016 NDHS used a multistage cluster sampling design which was stratified by geographical regions and urban-rural areas. The 2011 and 2016 NDHS were approved by the ethics committee of Nepal Health Research Council (NHRC) and human research ethics committee (HREC) in ICF Macro International. The Independent Review Boards of New Era and ICF Macro International reviewed and approved all the data collection tools and procedures for NDHS. This

study was based on an analysis of existing dataset in the DHS repository that are freely available online with all identifier information removed (http://dhsprogram.com). The first author communicated with MEASURE DHS/ ICF International, and permission was granted for the use of 2011 and 2016 NDHS.

A total of 25,536 women aged 15–49 years were interviewed in the two surveys (12,674 women in 2011 NDHS and 12,862 in 2016 NDHS) with an average response rate of over 97%. Survey data was collected from women with most recent live births 5 years prior each NDHS and women who were interviewed for the domestic violence module. As shown in S1 Fig, a total of 4079 and 3985 women had recent live births 5 years prior to 2011 and 2016 NDHS, respectively. Of which 1538 women were selected for the domestic violence module in 2011 NDHS and 1529 in 2016 NDHS. Combining both 2011 NDHS and 2016 NDHS, 3067 women who interviewed for domestic violence module had most recent live birth 5 years prior surveys. Women questionnaires designed for violence module were used to construct exposure variables. Details of the survey methodology, sampling procedures, and questionnaires are provided in the respective NDHS reports [17, 18].

## Dependent variable: Perinatal care

The dependent variable was perinatal care which is defined as care provided at 22 completed weeks of gestation till seven completed days after birth. Perinatal care takes into account antenatal and early postnatal care. WHO recommends a minimum of eight antenatal care contacts to reduce perinatal mortality and improve women's experience of care [2]. However, in this study 4 or more ANC visits was used because this was WHO recommendation at the time the 2011 and 2016 NDHS were conducted. For a positive pregnancy experience, 49 WHO recommendations on ANC were developed related to five types of interventions: (i) Nutritional interventions; (ii) Maternal and foetal assessment; (iii) Preventive measures; (iv) Interventions for common physiological symptoms; and (v) Health system interventions to improve utilization and quality of ANC [2]. In high mortality settings and where access to facility-based postnatal care is limited, WHO and UNICEF recommend at least two home postnatal visits for all home births: the first visit should occur within 24 hours from birth and the second visit on day 3 [20]. WHO guidelines for postnatal care addresses the timing, number, and place of postnatal contacts as well as the content of postnatal care for all mothers and babies during the six-week period after birth. For this study, we examined receipt of perinatal care using three indicators: (i) Non-use of 4 or more skilled ANC visits; (ii) Non-use of institutional delivery; and (iii) Non-use of skilled delivery assistants.

## Descriptive study variables

The potential confounding variables were type of residence (urban and rural), ecological region (Terai, hill or mountain), household wealth index (poor, middle, rich), maternal education (secondary and higher primary, no education), maternal current working status (currently not working and currently working), husband education (secondary and higher, primary, no education), ethnicity (Brahmin/Chettri, Janajati including newar, Dalit, others including Muslim), maternal age, parity, maternal smoking status, exposure to mass media and year of survey (NDHS 2011 and NDHS 2016). The household wealth index is a composite index based on a household's ownership of selected assets, such as televisions and bicycles, materials used for housing construction and types of water access and sanitation facilities [27]. The exposure variables were husband/partner alcohol drinking and IPV which was represented in three forms: physical violence, emotional violence, and sexual violence.

**Statistical analysis.**   This study pooled data from NDHS 2011 and 2016 to increase sample size. Analyses were performed using Stata version 15.0 (StataCorp, College Station, TX, USA). The dependent variables were based on the uptake of the three perinatal care services and coded as '0' [if respondents reported 4 or more ANC visits, if the deliveries took place in a health facility, or if the deliveries were assisted by doctors, nurses, or midwives]; and '1' [if respondents did not report 4 or more ANC visits, if the deliveries did not take place in a health facility, or if the deliveries were not assisted by doctors, nurses, or midwives]. Frequency tabulation was performed to describe the characteristics of study population. The Taylor series linearization method was used in the surveys to estimate the confidence intervals (Cls) around prevalence estimates of IPV. Generalized linear latent and mixed models (gllamm) with the logit link and binomial family [28] that adjusted for cluster and violence specific sampling weights were used to examine the impact of each of the exposure variables on the uptake of perinatal care while taking confounding variables into account. After fitting multivariate logistic regression models, Hosmer-Lemeshow goodness-of-fit test was performed by using 'svylogitgof' command in stata which showed non-significant results (p>0.05) suggesting adequate fit. In addition, time dependent confounder (year of survey) was included in the regression analysis to ensure the results are similar over a 5-year interval.

## Results

### Characteristics of the sample

Table 1 shows the characteristics of the weighted study sample with live births who were interviewed for domestic violence module in Nepal (2011–2016). A total weighted sample of 2,727 women were interviewed for the domestic violence module in both surveys with 1374 (50.4%) in 2011 NDHS and 1353 (49.6%) in 2016 NDHS. Of these, 37.3% lived in urban areas while 62.7% lived in rural areas. The Terai, Hill and Mountain regions had 44.9%, 45.8% and 9.3% inhabitants, respectively. Approximately 45% were poor and 34.8% rich. While 38% reported no maternal education and 42.6% reported secondary and higher maternal education, 16.9% reported no husband education and 60.2% reported secondary and higher husband education. About 43.2% mothers were not currently working while 56.8% were currently working. The sample also reported higher exposure to mass media (83.5%) and maternal smoking status of 7%. Husband/partner who drank alcohol was 47.6%, women who reported physical, emotional, and sexual violence were 22.3%, 14.1% and 11.4% respectively.

**Impact of violence.**   In the univariate analysis, women who were exposed to all three forms of violence: emotional, physical, and sexual violence were less likely to attend 4 or more skilled ANC visits, and utilize institutional delivery and skilled delivery assistants. While in the multivariable analysis, women who were exposed to physical violence were significantly less likely to utilize institutional delivery and skilled delivery assistants as shown in Table 2.

**Impact of male alcohol use.**   In the univariate analysis, women with partners who drank alcohol were less likely to attend 4 or more skilled ANC visits, and utilize institutional delivery and skilled delivery assistants than their counterparts with partners who do not drink alcohol as shown in Table 2.

**Impact of violence and male alcohol use (combined).**   In the univariate analysis, women who reported alcohol use with or without exposure to emotional violence were less likely to attend 4 or more skilled ANC visits, and utilize institutional delivery and skilled delivery assistants than their counterparts. While women who reported no alcohol use but exposure to emotional violence were less likely to utilize institutional delivery and skilled delivery assistants. Likewise, women who reported alcohol use with or without physical violence as well as women who reported no alcohol use, but exposure to physical violence were less likely to attend 4 or

**Table 1. Characteristics of study sample (weighted) with live births who were interviewed for domestic violence module in Nepal (2011–2016).**

| Characteristics | N | % |
|---|---|---|
| **Year of survey** | | |
| 2011 | 1374 | 50.4 |
| 2016 | 1353 | 49.6 |
| **Type of Residence** | | |
| Urban | 1018 | 37.3 |
| Rural | 1710 | 62.7 |
| **Ecological region** | | |
| Terai | 1225 | 44.9 |
| Hill | 1248 | 45.8 |
| Mountain | 255 | 9.3 |
| **Household wealth index** | | |
| Poor | 1228 | 45.0 |
| Middle | 550 | 20.2 |
| Rich | 950 | 34.8 |
| **Maternal education** | | |
| Secondary and higher | 1162 | 42.6 |
| Primary | 530 | 19.4 |
| No education | 1036 | 38.0 |
| **Maternal current working status** | | |
| Currently not working | 1178 | 43.2 |
| Currently working | 1550 | 56.8 |
| **Husband education(N = 2715)** | | |
| Secondary and higher | 1641 | 60.2 |
| Primary | 613 | 22.5 |
| No education | 461 | 16.9 |
| **Ethnicity** | | |
| Brahmin/Chettri | 871 | 31.9 |
| Janajati including newar | 902 | 33.1 |
| Dalit | 469 | 17.2 |
| Others including Muslim | 487 | 17.9 |
| **Maternal age** | | |
| 14–24 | 1150 | 42.2 |
| 25–34 | 1288 | 47.2 |
| 35–49 | 290 | 10.6 |
| **Parity** | | |
| 1 | 949 | 34.8 |
| 2 | 795 | 29.1 |
| 3+ | 985 | 36.1 |
| **Maternal smoking status** | | |
| No | 2536 | 93.0 |
| Yes | 191 | 7.0 |
| **Exposure to mass media** | | |
| No | 450 | 16.5 |
| Yes | 2278 | 83.5 |
| **Husband/partner drinking alcohol** | | |
| No | 1429 | 52.4 |

(*Continued*)

**Table 1.** (Continued)

| Characteristics | N | % |
|---|---|---|
| Yes | 1299 | 47.6 |
| **Physical violence** | | |
| No | 2118 | 77.2 |
| Yes | 609 | 22.3 |
| **Emotional violence** | | |
| No | 2343 | 85.9 |
| Yes | 385 | 14.1 |
| **Sexual violence** | | |
| No | 2416 | 88.6 |
| Yes | 312 | 11.4 |
| **Total** | **2728** | **100.0** |

more skilled ANC visits, and utilize institutional delivery than their counterparts. Furthermore, women who reported alcohol use with or without exposure to sexual violence were less likely to attend 4 or more skilled ANC visits, and utilize institutional delivery and skilled delivery assistants than their counterparts.

In the multivariable analysis, women who reported a combination of alcohol use by partner and exposure to emotional violence as well as women who reported a combination of alcohol use by partner and exposure to physical violence were less likely to attend 4 or more skilled ANC visits than their counterpart with no partner who drinks alcohol and no exposure to emotional or physical violence respectively. Women who reported no alcohol use by partner, but exposure to physical violence were less likely to utilize institutional delivery and skilled delivery assistants.

## Discussion

This study examined the association between IPV and male alcohol use and the receipt of perinatal care in Nepal. IPV and male alcohol use are known to expose women to a wide range of health problems due to complications from not accessing and utilising maternity health-care services [5, 17]. Over the study period (2011–2016), IPV and male alcohol use were reported to have a significant impact on the usage of maternity health-care services in Nepal.

In this study, women exposed to physical violence were less likely to access and utilise skilled delivery assistants. Research has shown that delivery attended by skilled professionals contribute to better pregnancy and childbirth outcomes, early detection, and management of complications in the ANC period, as well as during delivery and in the postnatal period [2, 29, 30]. The lack of utilisation of skilled birth attendants by women experiencing IPV could be due to sociocultural factors which prevent engagement with maternity health-care services. In line with our finding, a previous study found that there was a decrease in the use of maternity health-care services by women experiencing IPV due to poor education and health-seeking behaviour, coupled with a lack of adequate information [31]. In addition, a similar study conducted in Kenya also reported that women's experience of IPV may influence and subsequently reduce their usage of skilled birth attendance [32].

This study also found that women who are exposed to physical violence were less likely to utilise institutional delivery. This could be attributed to the need to maintain privacy and confidentiality. One of the factors that mitigate against women who experience IPV is the idea that family problems should be kept private [33]. These norms and values may act to

**Table 2. Impact of alcohol use and different forms IPV on receipt of perinatal care in Nepal (20011–2016).**

| Exposure variables | Non-use of 4 or more skilled ANC visits | | Non-use of institutional delivery | | Non-use of skilled delivery assistants | |
|---|---|---|---|---|---|---|
| | Unadjusted OR (95% CI) | Adjusted OR (95% CI) | Unadjusted OR (95% CI) | Adjusted OR (95% CI) | Unadjusted OR (95% CI) | Adjusted OR (95% CI) |
| **Emotional violence** | | | | | | |
| No | 1.00 | 1.00 | 1.00 | 1.00 | 1.00 | 1.00 |
| Yes | 1.73(1.32, 2.26) ** | 1.17(0.89, 1.55) | 1.53(1.16, 2.01) * | 1.07(0.80, 1.44) | 1.62(1.22, 2.16) * | 1.14(0.84, 1.54) |
| **Physical violence** | | | | | | |
| No | 1.00 | 1.00 | 1.00 | 1.00 | 1.00 | 1.00 |
| Yes | 1.87(1.48, 2.35) ** | 1.22(0.95, 1.55) | 1.94(1.52, 2.46) ** | 1.30(1.01, 1.68) * | 2.12(1.65, 2.74) ** | 1.43(1.10, 1.88) * |
| **Sexual violence** | | | | | | |
| No | 1.00 | 1.00 | 1.00 | 1.00 | 1.00 | 1.00 |
| Yes | 1.56(1.16, 2.10) * | 1.05(0.77, 1.44) | 1.45(1.06, 1.97) * | 0.98(0.71(1.36) | 1.55(1.13, 2.14) * | 1.00(0.71, 1.41) |
| **Alcohol use by male partner** | | | | | | |
| No | 1.00 | 1.00 | 1.00 | 1.00 | 1.00 | 1.00 |
| Yes | 1.74(1.44, 2.10) ** | 1.22(1.99, 1.50) | 1.53(1.26, 1.84) ** | 1.01(0.82, 1.25) | 1.50(1.23, 1.84) ** | 0.96(0.77, 1.19) |
| **Combination of alcohol use and emotional violence** | | | | | | |
| No alcohol no violence | 1.00 | 1.00 | 1.00 | 1.00 | 1.00 | 1.00 |
| Alcohol but no violence | 1.63(1.33, 1.99) ** | 1.17(0.95, 1.46) | 1.60(1.29, 1.98) ** | 1.10(0.88, 1.38) | 1.50(1.20, 1.87) ** | 0.99(0.79, 1.25) |
| No alcohol but violence | 1.41(0.89, 2.24) | 0.97(0.60, 1.57) | 2.14(1.33, 3.48) * | 1.67(0.98, 2.73) | 1.94(1.19, 3.14) ** | 1.48(0.90, 2.46) |
| Alcohol and violence | 2.60(1.87, 3.61) ** | 1.42(1.01, 2.00) * | 1.75(1.25, 2.45) ** | 0.91(0.63, 1.31) | 1.94(1.35, 2.76) ** | 0.99(0.68, 1.45) |
| **Combination of alcohol use and physical violence** | | | | | | |
| No alcohol no violence | 1.00 | 1.00 | 1.00 | 1.00 | 1.00 | 1.00 |
| Alcohol but no violence | 1.65(1.33, 2.04) ** | 1.20(0.96, 1.51) | 1.54(1.23, 1.92) ** | 1.06(0.83, 1.35) | 2.05(1.50, 2.80) ** | 0.97(0.76, 1.24) |
| No alcohol but violence | 1.83(1.24, 2.68) * | 1.19(0.81, 1.75) | 2.45(1.68, 3.58) ** | 1.67(1.13, 2.48) * | 1.42(0.68, 2.49) | 1.73(1.15, 2.61) * |
| Alcohol and violence | 2.60(1.95, 3.47) ** | 1.39(1.03, 1.88) * | 2.27(1.68, 3.07) ** | 1.17(0.85, 1.62) | 2.90(1.77, 4.75) ** | 1.26(0.90, 1.77) |
| **Combination of alcohol use and sexual violence** | | | | | | |
| No alcohol no violence | 1.00 | 1.00 | 1.00 | 1.00 | 1.00 | 1.00 |
| Alcohol but no violence | 1.73(1.41, 2.11) ** | 1.23(0.99, 1.52) | 1.47(1.19, 1.81) ** | 0.98(0.79, 1.23) | 1.39(1.12, 1.73) * | 0.91(0.72, 1.14) |
| No alcohol but violence | 1.59(0.98, 2.58) | 1.05(0.64, 1.74) | 1.20(0.73, 1.97) | 0.83(0.50, 1.40) | 1.08(0.64, 1.81) | 0.71(0.42, 1.22) |
| Alcohol and violence | 2.31(1.59, 3.36) ** | 1.23(0.83, 1.81) | 2.11(1.43, 3.12) ** | 1.07(0.70, 1.63) | 2.40(1.59, 3.62) ** | 1.15(0.74, 1.79) |

*P Value<0.05;

**P Value<0.001;

Model adjusted for Year of survey, type of residence, ecological region, household wealth index, maternal education, maternal current working status, husband/partner education, ethnicity, maternal age, parity, maternal working status, and exposure to mass media.

undermine the impact of IPV and deter women from seeking help and support. Furthermore, women experiencing physical violence tend to avoid circumstances that would result in involuntary reporting of their spouses for fear of continued abuse. These women also avoid situations where they are forced to interact with institutions that are mandatory reporters, including hospitals. Universal screenings administered by health or social work professionals are being carried out in antenatal and other health care settings to detect domestic and family violence and mandatory reports are made [34].

In this study, women who reported experiencing emotional or physical violence with a partner who consume alcohol were less likely to attend the recommended antenatal visits. Alcohol abuse by men has been found to be a significant determinant of IPV against women [35–37]

and increase the display of controlling behaviours by the man such as demanding the woman ask for permission before seeking healthcare [38]. This increased display of controlling behaviours and violence might be due to an impaired cognitive function, which reduces self-control and leaves the individual less capable of negotiating a non-domineering and non-violent resolution to conflicts. A report by the WHO also identifies alcohol consumption at harmful levels as a major contributor to the occurrence of IPV [17]. The impact of such behaviour may lead to partner interference with recommended ANC visits attendance. A similar study carried out in Mozambique [39] found that emotional violence was the most reported form of partner violence which impacts on a woman's utilisation of ANC service. The study also reported that women experiencing emotional and physical abuse women were more likely to initiate and access ANC late during pregnancy resulting in possible high mortality [40].

This study had some limitations. First, the analyses were based on cross-sectional data, hence, causality cannot be established between study outcome and confounding factors. Second, despite the use of a comprehensive set of variables in our analysis, the effect of residual confounding as a result of unmeasured co-variates which might influence the receipt of perinatal care, such as pregnancy complications, pre-existing maternal health conditions, quality of maternal care available as well as personal or sociocultural perceptions may have been ruled out. However, this study also had several strengths. First, the 2011 and 2016 NDHS were nationally representative survey which used standardized methods that achieved an average response rate of 97%; therefore, the findings from this study could be generalized to the entire Nepalese population and is unlikely to be affected by selection bias. Second, this study used pooled data from 2011 and 2016 NDHS with large sample size and increased statistical power. Third, the use of random effect multilevel modelling which accounts for the hierarchical structure of the data and the variability within the exploratory variables better estimates the level of association between the potential confounding factors with the study outcome [41]. Finally, standardized uniform questionnaires, were used to collect information across both surveys which increase accuracy and promote coherence of the data used for analysis.

This study is useful in public health planning to reinforce the need to support women experiencing IPV through the pregnancy continuum in accessing perinatal care. It will also assist the Nepalese government in developing and implementing appropriate programs aimed at improving receipt of perinatal care amongst vulnerable women.

## Conclusions

This research aimed to fill a gap in the literature and explored the association between IPV and male alcohol use and perinatal care seeking in Nepal. The statistical analysis have shown that women who experienced IPV either in the form of physical or emotional abuse and reported alcohol use by partner were less likely to receive the WHO recommended level of perinatal care, which is attend 4 or more antenatal clinics run by trained health care providers. This reduction in antenatal care prevents women from receiving the care and treatment required for preventing maternal deaths and promoting mother and child wellbeing. Moreover, the negative association between IPV and perinatal care, suggests there are social, cultural and structural barriers that prevent these women from accessing crucially needed care not just for their pregnancy but also for addressing their mental health needs arising from their IPV experience. Thus, it is essential to develop comprehensive interventions addressing not just the interpersonal level factors between the couple, and mothers-in-law, but also the social cultural norms that perpetuate IPV and alcohol misuse and prevent women from seeking perinatal care. This calls for community-based interventions including screening for IPV designed in a way that makes it easier for women to get the help they need. Integrating perinatal, mental

health and IPV support services with other health services and providing a home visiting service may contribute to increasing the uptake of perinatal care and reducing the incidence of IPV.

## Supporting information

**S1 Fig. Composition of study sample (unweighted numbers).**
(DOCX)

## Author Contributions

**Conceptualization:** Blessing Akombi-Inyang, Pramesh Raj Ghimire, Emma Woolley, Husna Razee.

**Formal analysis:** Blessing Akombi-Inyang, Pramesh Raj Ghimire.

**Methodology:** Blessing Akombi-Inyang, Pramesh Raj Ghimire, Elizabeth Archibong.

**Writing – original draft:** Blessing Akombi-Inyang.

**Writing – review & editing:** Blessing Akombi-Inyang, Pramesh Raj Ghimire, Elizabeth Archibong, Emma Woolley, Husna Razee.

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
