## [Decision Letter · Decision Letter 0]

18 Aug 2021

PONE-D-21-11367

Association between intimate partner violence and male alcohol use and the receipt of perinatal care Evidence from Nepal demographic and health survey 2001–2016

PLOS ONE

Dear Dr. Akombi-Inyang,

Thank you for submitting your manuscript to PLOS ONE. After careful consideration, we feel that it has merit but does not fully meet PLOS ONE’s publication criteria as it currently stands. Therefore, we invite you to submit a revised version of the manuscript that addresses the points raised during the review process.

We look forward to receiving your revised manuscript.

Kind regards,

Sandi Dheensa

Academic Editor

PLOS ONE

Journal Requirements:

**Additional Editor Comments (if provided):**

As well as addressing reviewer comments, please address the following:

- PLOS ONE has a publication critiera that says 'studies involving humans categorized by race/ethnicity, age, disease/disabilities, religion, sex/gender, sexual orientation, or other socially constructed groupings, authors should explicitly describe their methods of categorizing human populations'. Given this, please can you explain why you use the following groupings for ethnicity? Brahmin/Chettri, Janajati including newar, Dalit, others including Muslim. Is this a standard way to categorise ethnicities in Nepal?

- Please describe the contributions of all authors - see https://journals.plos.org/plosone/s/authorship for guidance

- Please use the apropriate reporting guidelines to report your work - see here: https://journals.plos.org/plosone/s/submission-guidelines The STROBE cross sectional checklist might be the most appropriate for you

- Please rephrase the following sentence 'However, the role of male alcohol consumption in shaping the extent of IPV as well as its influence on the receipt of perinatal care cannot be over-emphasized' - this sentence does not make sense in the current place in the manuscript as you have not yet shown what the role of alcohol is.

Reviewers' comments:

Reviewer's Responses to Questions

**Comments to the Author**

1. Is the manuscript technically sound, and do the data support the conclusions?

Reviewer #1: Partly

Reviewer #2: Yes

Reviewer #3: No

2. Has the statistical analysis been performed appropriately and rigorously? 

Reviewer #1: No

Reviewer #2: No

Reviewer #3: Yes

3. Have the authors made all data underlying the findings in their manuscript fully available?

Reviewer #1: Yes

Reviewer #2: Yes

Reviewer #3: Yes

4. Is the manuscript presented in an intelligible fashion and written in standard English?

Reviewer #1: Yes

Reviewer #2: Yes

Reviewer #3: Yes

5. Review Comments to the Author

**Reviewer #1**

This is a manuscript on secondary data analysis, exploring the association of intimate partner violence (IPV) and male alcohol use on the receipt of perinatal care. The study objective is timely, and on target. I have some comments mostly on the statistical analysis side:

(a) The dependent variable "perinatal care" has been explained in a separate section, however, it was not made clear the exact nature of the response, i.e., is it a binary response (which I think it is), or something else. If binary, clearly mention how the cutoff was determined. It is not clear.

(b) Taylor series linearization method was employed to calculate the variance estimates; some more details needed for the reader. Is it automatic inside Stata?

(c) Please explain why both logit and binomial family was used under the Generalized linear latent and mixed modeling. Where is the mixed modeling coming from (like, what is the cluster)? Be clear. What is the context of binomial modeling?

(d) No goodness-of-fit measures were provided; see below (STATA may have such a command)

https://www.stata-journal.com/article.html?article=st0099

(e) A small sample size/power description may allow readers to understand what effect size the authors wanted to achieve before conducting the analysis, and the appropriateness of the sample size. You may ignore the survey weighting while calculating the power, if such a program, or method doesn't exist.

**Reviewer #2: **

This paper used the 2011 and 2016 Nepal Demographic and Health Surveys to look at the association between IPV, male alcohol use and perinatal care. The introduction should be reorganized to flow better and focus on the context of Nepal. The paper can be strengthened by expanding upon why the analysis methods were chosen.

**Reviewer #3: **

The 2011 and 2016 NDHS data is used together and analysed. But it was not clear that both instruments and methods were the same. It should be explained in methods.

Analysis

The 2011 and 2016 NDHS data were combined and analysed. No effort was taken to show that both samples and findings are similar. It is difficult to assume that the factors are unchanged over a 5-year interval and here it looks like the two different data sets are combined and analysed to get results. the combination of two data sets seems to be purely for the increase in sample size without considering any other factors.

An option is to compare and contrast the data set before combining both. In this combination, authors can compare the similarities of key factors such as IPV, alcohol consumption and ANC care. If both data sets are similar, they can be combined and analysed. if not it's good to analyze it separately and compare.

6. PLOS authors have the option to publish the peer review history of their article (what does this mean?). If published, this will include your full peer review and any attached files.

Reviewer #1: No

Reviewer #2: No

Reviewer #3: **Yes: **Muzrif Munas

---

## [Author Response · Author response to Decision Letter 0]

29 Sep 2021

Editor comments 

Comment: PLOS ONE has a publication criterion that says 'studies involving humans categorized by race/ethnicity, age, disease/disabilities, religion, sex/gender, sexual orientation, or other socially constructed groupings, authors should explicitly describe their methods of categorizing human populations'. Given this, please can you explain why you use the following groupings for ethnicity? Brahmin/Chettri, Janajati including newar, Dalit, others including Muslim. Is this a standard way to categorise ethnicities in Nepal?

Response: Thank you for your comment. The listed groupings for ethnicity in the study are the recognised ethnic groups in Nepal and are reported as such in all Nepalese Demography and Health Surveys (2006, 2011 and 2016). 

In Nepal, Brahmin/Chettri is considered as advantaged ethnic group and belongs to the top of ethnic hierarchy whereas Janajati including Newar, and Dalit are relatively disadvantaged and socioeconomically marginalized groups. Others including Muslim are different group of people who are neither Brahmin/Chettri nor Dalit or Janajati and has been categorised as others including Muslim. Some of these have been discussed in the final report of NDHS (2011-2016). Please see https://dhsprogram.com/pubs/pdf/FR336/FR336.pdf

Comment: Please describe the contributions of all authors – see https://journals.plos.org/plosone/s/authorship for guidance

Response: Thank you for your advice. However, the document you shared provides information on PLOS ONE’s authorship policies and does not state that the information should be included in the manuscript. Furthermore, following PLOS ONE’s submission guidelines (https://journals.plos.org/plosone/s/submission-guidelines#loc-style-and-format), there is no section within the manuscript where authors’ contribution is solicited. However, we have clearly indicated each authors’ contribution in the submission system as advised in PLOS ONE’s authorship policy.

Comment: Please use the appropriate reporting guidelines to report your work - see here: https://journals.plos.org/plosone/s/submission-guidelines The STROBE cross sectional checklist might be the most appropriate for you

Response: Thank you for your advice. We adhered strictly to PLOS ONE’s submission guidelines as outlined in https://journals.plos.org/plosone/s/submission-guidelines. We also adhered to STROBE cross sectional study checklist to the extent suitable for our study given that our study is based on secondary data analysis. 

Comment: Please rephrase the following sentence 'However, the role of male alcohol consumption in shaping the extent of IPV as well as its influence on the receipt of perinatal care cannot be over-emphasized' - this sentence does not make sense in the current place in the manuscript as you have not yet shown what the role of alcohol is.

Response: Thank you for your insightful comment. We have considered your comment and revised the manuscript accordingly. The manuscript now reads 

“However, the role of male alcohol consumption in shaping the extent of IPV as well as its influence on the receipt of perinatal care is still largely uncertain”. Please see lines 127 - 129.

Reviewer #1 comment

This is a manuscript on secondary data analysis, exploring the association of intimate partner violence (IPV) and male alcohol use on the receipt of perinatal care. The study objective is timely, and on target. I have some comments mostly on the statistical analysis side:

Comment: (a) The dependent variable "perinatal care" has been explained in a separate section, however, it was not made clear the exact nature of the response, i.e., is it a binary response (which I think it is), or something else. If binary, clearly mention how the cutoff was determined. It is not clear.

Response: Thank you for your comment. We have revised the manuscript to make clear the nature of the response as follows: 

“The dependent variables of this study were based on the uptake of the three perinatal care services and coded as ‘0’ [if respondents reported 4 or more ANC visits, if the deliveries took place in a health facility, or if the deliveries were assisted by doctors, nurses, or midwives]; and ‘1’ [if respondents did not report 4 or more ANC visits, if the deliveries did not take place in a health facility, or if the deliveries were not assisted by doctors, nurses, or midwives]”. Please see lines 215 – 220. 

Comment: (b) Taylor series linearization method was employed to calculate the variance estimates; some more details needed for the reader. Is it automatic inside Stata?

Response: Thank you for your comment. However, we believe that the information provided on the Taylor series linearization method is sufficient to provide the reader with the necessary details to validate our analysis.

Comment: (c) Please explain why both logit and binomial family was used under the Generalized linear latent and mixed modeling. Where is the mixed modeling coming from (like, what is the cluster)? Be clear. What is the context of binomial modeling?

Response: First of all, we would like to highlight that when the link function is the logit function, the binomial regression becomes the well-known logistic regression. Both the logit and binomial family was used because Generalized linear latent and Mixed Models (GLLAMM) is a stata program to fit multilevel latent variable models for (multivariate) responses of mixed type including counts, duration, ordered and unordered categorical responses [Please see: Rabe-Hesketh S, Skrondal A. Multilevel and longitudinal modeling using Stata. STATA press; 2008]. To use GLLAMM program in stata, it is mandatory to first identify the link and corresponding family; and therefore, both the logit and corresponding family (binomial) were used. In the past, we have extensively used this program to fit multivariate regression models that also adjust for multi-stage clustering and sampling weights for complex survey design like DHS [Please see: https://link.springer.com/article/10.1186/s12884-019-2234-6;
https://journals.plos.org/plosone/article?id=10.1371/journal.pone.0202603

Comment: (d) No goodness-of-fit measures were provided; see below (STATA may have such a command)

https://www.stata-journal.com/article.html?article=st0099

Response: Thank you for the important comment, and we have performed Hosmer-Lemeshow goodness-of-fit test [Please see https://journals.sagepub.com/doi/10.1177/1536867X0600600106], and the text below has been added into the revised manuscript. 

“After fitting multivariate logistic regression models, Hosmer-Lemeshow goodness-of-fit test was performed by using ‘svylogitgof’ command in stata which showed non-significant results (p>0.05) suggesting adequate fit”. Please see lines 226 – 228.

Comment: (e) A small sample size/power description may allow readers to understand what effect size the authors wanted to achieve before conducting the analysis, and the appropriateness of the sample size. You may ignore the survey weighting while calculating the power, if such a program, or method doesn't exist.

Response: This study was based on secondary data analysis of 2011 and 2016 NDHS which were nationally representative surveys with an average response rate of 97%. The sample size has already been calculated for each DHS (Please see individual DHS report for NDHS 2011 and 2016). However, we adjusted for difference in cluster and survey weight.

Reviewer #2 comment

This paper used the 2011 and 2016 Nepal Demographic and Health Surveys to look at the association between IPV, male alcohol use and perinatal care. The introduction should be reorganized to flow better and focus on the context of Nepal. The paper can be strengthened by expanding upon why the analysis methods were chosen.

Comment: 1. Further description on factors that affect maternal mortality for women in Nepal specifically will be helpful. 

Response: Thank you for your comment. Our study is focused primarily on perinatal care in Nepal and as such our narrative is in-line with the direction of our study. We recognize the impact perinatal care has on maternal mortality and has briefly acknowledged that. However, further discussion on the factors that affect maternal mortality for women in Nepal is out of the scope of our study.

Comment: 2. It would be useful to outline other pathways by which IPV reduces uptake of perinatal care services outside from decision making and poor mental health. 

Response: Thank you for your comment. To the extent of our literature review and based on WHO report, we have identified and hence acknowledge these pathways through which IPV reduces uptake of perinatal care services. As we identify more pathways, we will be including them in our upcoming studies. 

Comment: 3. Make sure there is consistency in using the term perinatal care and antenatal care. 

Response: Thank you for your comment. We have included a definition of perinatal care and antenatal care through which we believe will address any ambiguity in the use of the terms in our manuscript. 

The manuscript now reads: 

“Within the continuum of care, perinatal care is the care given to women from pregnancy through to one year after childbirth with antenatal care and postnatal care being care given before and after childbirth respectively”. Please see lines 80-82.

Comment: 4. The statement about mothers-in-law and their role seems misplaced. Either expand upon their role and how this relates to IPV or take it out. 

Response: Thank you for your comment. We agree that the statement seems misplaced and have taken it out.

Comment: 5. Why are only 4 or more ANC visits considered use of ANC when the recommendation is 8? 

Response: Thank you for your comment. Though WHO recommends 8 ANC visits, Nepal still uphold 4 ANC visits as the national standard, and this is what is reported in Nepal Demography and Health Surveys (DHS). The manuscript has been revised to reflect this information. The manuscript now reads:

In Nepal, standard ANC services include at least four ANC visits (first at the fourth month, second at the sixth month, third at the eighth month, and fourth at the ninth month of pregnancy). Please see lines 92 – 94.

Comment: 6. Clarification and a reason as to why this analysis plan was chosen would strengthen the statistical methods. 

Response: We do not quite understand this comment. We believe that the analysis is self-explanatory. First, we estimated prevalence, and their 95% CI to understand the characteristics of study sample, followed by fitting multivariate logistic regression model while taking into account cluster and sampling weights (specific to violence module) to understand the impact of male alcohol use and IPV on the uptake of perinatal care services in Nepal. 

Comment: 1. Line 104-105 in the introduction needs a citation. 

Response: Thank you for your comment. A citation has now been added. Please line 112.

Comment: 2. Line 124-127 needs a citation. 

Response: Thank you for your comment. Unfortunately, we could not find a suitable citation for the suggested statement. Therefore, we have removed the statement from the manuscript. 

Comment: 3. Line 139 needs a citation. 

Response: Thank you for your comment. A citation has now been added. Please see line 139.

Reviewer #3 comment

Comment: The 2011 and 2016 NDHS data is used together and analysed. But it was not clear that both instruments and methods were the same. It should be explained in methods.

Response: Thank you for your comment. In the method section we clearly stated that DHS including Nepal DHS use standardized questionnaires, manuals, field procedures and sampling techniques that are comparable across countries and time. Please see lines 162 – 166. 

Comment: The 2011 and 2016 NDHS data were combined and analysed. No effort was taken to show that both samples and findings are similar. It is difficult to assume that the factors are unchanged over a 5-year interval and here it looks like the two different data sets are combined and analysed to get results. the combination of two data sets seems to be purely for the increase in sample size without considering any other factors.

An option is to compare and contrast the data set before combining both. In this combination, authors can compare the similarities of key factors such as IPV, alcohol consumption and ANC care. If both data sets are similar, they can be combined and analysed. if not it's good to analyze it separately and compare.

Response: This is a valid point indeed. However, the aim of this study was not to compare and contrast key factors between the two NDHS (NDHS 2011 and NDHS 2016), rather was to provide a stronger evidence based on a pooled dataset that also takes time dependent confounder (year of survey) into account. The inclusion of time dependent confounder (year of survey) in the regression analysis is to ensure our results are similar over a 5-year interval. We have clarified the inclusion of time dependent confounder (year of survey) in our method section of the revised manuscript.

---

## [Decision Letter · Decision Letter 1]

22 Oct 2021

PONE-D-21-11367R1Association between intimate partner violence and male alcohol use and the receipt of perinatal care Evidence from Nepal demographic and health survey 2011–2016PLOS ONE

Dear Dr. Akombi-Inyang,

Thank you for submitting your manuscript to PLOS ONE. After careful consideration, we feel that it has merit but does not fully meet PLOS ONE’s publication criteria as it currently stands. Therefore, we invite you to submit a revised version of the manuscript that addresses the points raised during the review process.

We look forward to receiving your revised manuscript.

Kind regards,

Sandi Dheensa

Academic Editor

PLOS ONE

Journal Requirements:

Reviewers' comments:

Reviewer's Responses to Questions

**Comments to the Author**

1. If the authors have adequately addressed your comments raised in a previous round of review and you feel that this manuscript is now acceptable for publication, you may indicate that here to bypass the “Comments to the Author” section, enter your conflict of interest statement in the “Confidential to Editor” section, and submit your "Accept" recommendation.

Reviewer #1: All comments have been addressed

Reviewer #2: (No Response)

2. Is the manuscript technically sound, and do the data support the conclusions?

Reviewer #1: (No Response)

Reviewer #2: Yes

3. Has the statistical analysis been performed appropriately and rigorously? 

Reviewer #1: (No Response)

Reviewer #2: Yes

4. Have the authors made all data underlying the findings in their manuscript fully available?

Reviewer #1: (No Response)

Reviewer #2: Yes

5. Is the manuscript presented in an intelligible fashion and written in standard English?

Reviewer #1: (No Response)

Reviewer #2: Yes

6. Review Comments to the Author

Reviewer #1: (No Response)

Reviewer #2: 1. Though it’s helpful to have the code in the statistical analysis section, it might be better be better placed in a table or annex rather than in the text.

2. What assumptions, if any, were made about the data in order to perform the statistical analysis?

3. Can you further explain how husband alcohol use was quantified (ie, if binary is it that the husband could range from drinking one drink to many in a day)? In the discussion the focus is on alcohol abuse by men, might me good to add in studies that focus on recreational alcohol use as well if the variable of husband alcohol use ranges from recreational use to abuse.

4. Might be important to talk about stigma about reporting IPV and how that could effect your results.

7. PLOS authors have the option to publish the peer review history of their article (what does this mean?). If published, this will include your full peer review and any attached files.

Reviewer #1: No

Reviewer #2: No

---

## [Author Response · Author response to Decision Letter 1]

25 Oct 2021

Response to Reviewer 2

Comment 1: Though it’s helpful to have the code in the statistical analysis section, it might be better be better placed in a table or annex rather than in the text.

Response: Thank you for your suggestion. However, after reviewing several similar publications we are of the opinion that the preferred way to ensure our study is well understood by the reader is to briefly include the coding in the statistical analysis section. We have also published numerous studies in PLOS ONE as well as in other journals giving the coding information in the statistical analysis section. Please see some similar publications below:

https://journals.plos.org/plosone/article?id=10.1371/journal.pone.0236435

https://journals.plos.org/plosone/article?id=10.1371/journal.pone.0202603

https://journals.plos.org/plosone/article?id=10.1371/journal.pone.0223385

https://journals.plos.org/plosone/article?id=10.1371/journal.pone.0203278

https://www.hindawi.com/journals/bmri/2020/5487164/

Comment 2: What assumptions, if any, were made about the data in order to perform the statistical analysis?

Response: Thank you for your comment. However, we do not fully understand your comment. Our study was based on secondary data analysis, and we have provided all necessary information in the statistical analysis section. All limitations to the study have also been stated in the limitation section.

Comment 3: Can you further explain how husband alcohol use was quantified (ie, if binary is it that the husband could range from drinking one drink to many in a day)? In the discussion the focus is on alcohol abuse by men, might me good to add in studies that focus on recreational alcohol use as well if the variable of husband alcohol use ranges from recreational use to abuse.

Response: In NDHS (2011-2016), women completing the violence module were asked - Does (did) your (last) (husband/partner) drink alcohol? and the results were recorded as binary (YES/NO). Data on quantity of alcohol consumption were not collected in NDHS. Therefore, this study is not able to distinguish if alcohol use by husband was recreational or abuse. 

Comment 4: Might be important to talk about stigma about reporting IPV and how that could effect your results.

Response: Thank you for your comment. In the limitation section, we include "personal or sociocultural perceptions" as an unmeasured co-variate which might have been ruled out in the analysis. Stigma, we believe falls under that category. In addition, further discussion on stigma and how it impact on IPV is beyond the scope of our study.

---

## [Decision Letter · Decision Letter 2]

2 Nov 2021

Association between intimate partner violence and male alcohol use and the receipt of perinatal care Evidence from Nepal demographic and health survey 2011–2016

PONE-D-21-11367R2

Dear Dr. Akombi-Inyang,

We’re pleased to inform you that your manuscript has been judged scientifically suitable for publication and will be formally accepted for publication once it meets all outstanding technical requirements.

Kind regards,

Sandi Dheensa

Academic Editor

PLOS ONE

Additional Editor Comments (optional):

Reviewers' comments:

Reviewer's Responses to Questions

**Comments to the Author**

1. If the authors have adequately addressed your comments raised in a previous round of review and you feel that this manuscript is now acceptable for publication, you may indicate that here to bypass the “Comments to the Author” section, enter your conflict of interest statement in the “Confidential to Editor” section, and submit your "Accept" recommendation.

Reviewer #1: All comments have been addressed

Reviewer #2: All comments have been addressed

2. Is the manuscript technically sound, and do the data support the conclusions?

Reviewer #1: (No Response)

Reviewer #2: Yes

3. Has the statistical analysis been performed appropriately and rigorously? 

Reviewer #1: (No Response)

Reviewer #2: Yes

4. Have the authors made all data underlying the findings in their manuscript fully available?

Reviewer #1: (No Response)

Reviewer #2: Yes

5. Is the manuscript presented in an intelligible fashion and written in standard English?

Reviewer #1: (No Response)

Reviewer #2: Yes

6. Review Comments to the Author

Reviewer #1: (No Response)

Reviewer #2: (No Response)

7. PLOS authors have the option to publish the peer review history of their article (what does this mean?). If published, this will include your full peer review and any attached files.

Reviewer #1: No

Reviewer #2: No

---

## [Editor Report · Acceptance letter]

18 Nov 2021

PONE-D-21-11367R2 

Association between intimate partner violence and male alcohol use and the receipt of perinatal care: Evidence from Nepal demographic and health survey 2011–2016 

Dear Dr. Akombi-Inyang:

I'm pleased to inform you that your manuscript has been deemed suitable for publication in PLOS ONE. Congratulations! Your manuscript is now with our production department. 

Kind regards, 

on behalf of

Dr. Sandi Dheensa 

Academic Editor

PLOS ONE